# A Novel Pseudo Label-based Unsupervised Multiple Target Domain Adaptation Framework for Abdominal Organ Segmentation

Yuntao Zhu[1(✉)][0000−0003−2816−2709], Liwen Zou[1][0000−0003−4085−4003], Pengxu Wen[1][0009−0000−5211−4876], Ziwei Nie[1], Luying Gui[2], and Xiaoping Yang[1]

[1] Department of Mathematics, Nanjing University, Nanjing, China
`YuntaoZhu7@smail.nju.edu.cn`
[2] Department of Mathematics, Nanjing University of Science and Technology, Nanjing, China

**Abstract.** Obtaining labeled data from medical images is very expensive and labor intensive. We consider the problem of unsupervised domain adaptation (UDA) in cross-modality medical image segmentation, aiming to perform segmentation on the unannotated target domain with the help of labeled source domain. We present a framework that combines unsupervised domain adaptation, registration, and pseudo-label learning to effectively and flexibly adapt to multiple target domains. We introduce a novel image translation method based on anatomy space and a novel operation of matching and registration to improve pseudo-labels, which effectively mitigates the large cross-modality domain gap. Experiments demonstrate that our method achieves the average Dice Similarity Coefficient of 0.6053 and Normalized Surface Dice of 0.6462 on 13 abdominal organ segmentation tasks. Moreover, it significantly improves the inference speed, with an average running time of 20.7 seconds, and uses only an average of 1107332.6 MB of total GPU memory on final testing set.

**Keywords:** Unsupervised domain adaptation · Accelerate inference · Lightweight network.

## 1   Introduction

Deep learning has revolutionized the field of medical image segmentation, enabling accurate segmentation of various anatomical structures and lesions [21]. However, well-trained deep models usually perform poorly in real scenarios due to the severe data distribution difference between training and test sets caused by different imaging modalities, scanning protocols, and/or demographic properties. Domain adaptation (DA) is a promising solution to address the problem of dramatic performance degradation across modalities at inference time. It attempts to establish a mapping between the source and target domains so that models trained in the source domain can perform well in the target domain. Among DA, unsupervised domain adaptation (UDA) is more attractive and feasible,

where the ground truth in the target domain is not required. Various methods have been proposed to deal with this problem by aligning the source and target domains in terms of image appearance, feature distribution, or output structure. CycleGAN [30] and Contrastive Unpaired Translation (CUT) [20], focus on aligning image appearance between the source and target domains. However, these methods may introduce distortions to the anatomical structure of the images, which can hinder accurate segmentation [26]. Long et al. [11] focused on alignment at the feature level. FPL+ [25] aim to align the domains at both the image and feature levels by using pseudo labels. However, an overly simplified assumption for style transfer, which is a commonly used strategy of image alignment, is adopted in most previous work, where they model this process as a deterministic one-to-one mapping with a mean style of each domain. In fact, image styles in a single domain may are quite different [28].

In this work, we propose a multiple target domain adaptation framework for UDA in 3D medical image segmentation. First, an Anatomy Space Image Translation (ASIT) using CycleGAN enables the alignment of CT and MRI images, which builds a bi-directionality mapping between CT and MRI. In this way, we can eliminate the impact of the domain shift on networks. Second, we can train a segmentation network using the fake MRI generated by CT and ground truth. Then, The trained network infers real MRI to gain pseudo-labels. Since the MRI data have multiple sequences, e.g. T1, T2, DWI, some sequences show better pseudo-label quality than others. We conduct a mixed operation (multi-domain matching, MDM) of matching and registration to improve pseudo-labels. Third, after we own an MRI dataset with high-quality pseudo-labels, we do self-training to enhance network performance by filtering some bad cases. Finally, we use the selected MRI dataset to train a small network and do a fast inference process, that can speed the inference and reduce computational resources.

The contributions of this work are summarized in three aspects:

- We propose a novel UDA framework 3D medical image segmentation based on generating high-quality pseudo labels in the multiple target domains.
- We introduce a novel image translation method (ASIT) based on Anatomy Space and a novel operation (MDM), which effectively mitigates the large cross-modality domain gap.
- The experiment shows that our method improves the ability of segmentation networks for the unsupervised domains adaptation. This outperforms the single target domain adaptation by 23 percentage points for the Dice Similarity Coefficient (DSC).

## 2   Method

As shown in Figure 1, our method has four stages: 1) Image translation using CycleGAN enables the alignment of CT and MRI images, which builds a bi-directionality mapping between CT and MRI. In this way, we can eliminate the impact of the domain shift on networks. 2) We can train a segmentation network using the fake MRI generated by CT and ground truth. Then, The

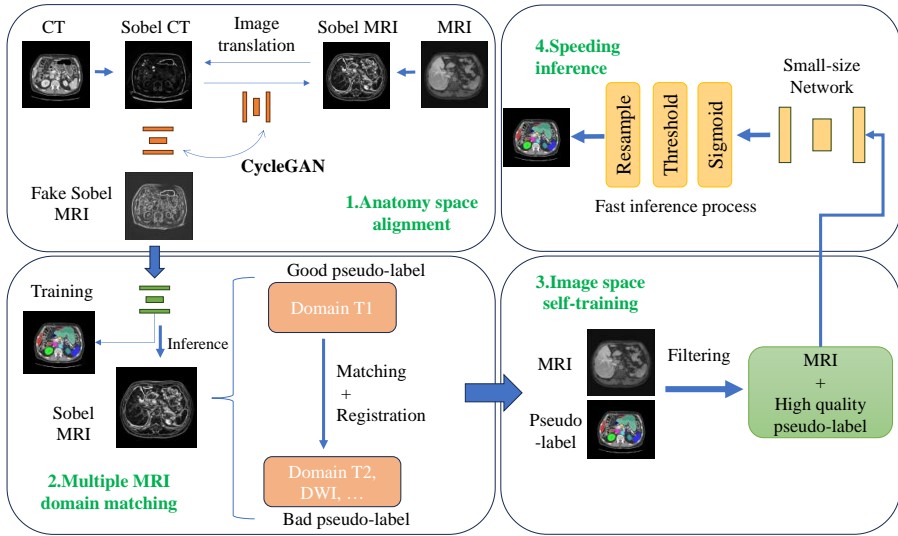

Fig. 1: The proposed method framework.

trained network infers real MRI to gain pseudo-labels. Since the MRI data have multiple sequences, e.g. T1, T2, DWI, some sequences show better pseudo-label quality than others. We conduct a mixed operation of match and registration to improve pseudo-labels. 3) After we own an MRI dataset with high-quality pseudo-labels, we do self-training to enhance network performance by filtering some bad cases. 4) Finally, we use the selected MRI dataset to train a small network and do a fast inference process, that can speed the inference and reduce computational resources. In addition, we used the pseudo labels generated by the FLARE22 winning algorithm [8] and the best-accuracy-algorithm [23] as labels of CT.

### 2.1   Anatomy space image translation

To achieve UDA for 3D medical image segmentation, our idea first obtains high-quality pseudo labels for training images in the target domain and then trains a segmentation model in that domain by learning from pseudo labels. We find an interesting fact that CT and MRI have consistent anatomic structures in the body. Undoubtedly, an organ's shape, such as the liver, kidneys, etc, does not change with the modal changing, it is an invariant feature. Utilizing this feature, we can build a network to transfer segmentation ability between various modalities.

Let $\mathcal{D}_s$ and $\mathcal{D}_t$ denote a set of labeled source-domain images and a set of unlabeled target-domain images, respectively. Let $X_i^s$ and $X_j^t$ denote the $i$-th image from $\mathcal{D}_s$ and the $j$-th image from $\mathcal{D}_t$, respectively, where the label of $X_i^s$ is $Y_i^s$. Note that the source domain and target domain are from different patient

groups, i.e., $X_i^s$ and $X_j^t$ are unpaired. Due to the domain shift between $\mathcal{D}_s$ and $\mathcal{D}_t$, training a model with $\mathcal{D}_s$ to generate pseudo labels for $\mathcal{D}_t$ will lead to a poor performance. To improve the quality of pseudo labels for $\mathcal{D}_t$, we propose Anatomy Space Image Translation (ASIT) to augment $\mathcal{D}_s$ before training the pseudo label generator.

Specifically, we utilize an image style translator $T_t$ to translate a labeled source-domain image $X_i^s$ into a pseudo target-domain image $X_i^{s \to t} = T_t(X_i^s)$, and use another image style translator $T_s$ to translate $X_i^{s \to t}$ back to the source domain, leading to a pseudo source-domain image $X_i^{s'} = T_s(X_i^{s \to t})$. Note that $T_t$ and $T_s$ are often trained jointly for learning from unpaired training sets, as used in CycleGAN [30]. As the training sets are unpaired, it is difficult to make $X_i^{s \to t}$ and $X_t^{s'}$ exactly match the ground truth target-modality and source-modality images, respectively. However, we employ the Sobel filter on CT and MRI first to obtain Sobel-CT and Sobel-MRI, the modality difference achieves a large degree of reduction. As illustrated in Figure 2, the Anatomy Space Image Translation shows a good performance.

In this work, the image translators $T_s$ and $T_t$ are implemented based on Cycle-GAN [30] with two discriminators $D_s$ and $D_t$ for the two domains, respectively. The training involves two adversarial losses $\mathcal{L}_{gan}^t$, $\mathcal{L}_{gan}^s$, a cycle consistency loss $\mathcal{L}_{cyc}$ and an identity mapping loss $\mathcal{L}_{\text{identity}}$. The target-domain adversarial loss $\mathcal{L}_{gan}^t$ is:

$$
\begin{aligned}
\mathcal{L}_{gan}^t(T_t, D_t) =& \mathbb{E}_{X_j^t \sim \mathcal{D}_t}[\log D_t(X_j^t)] \\
&+ \mathbb{E}_{X_i^s \sim \mathcal{D}_s}[\log(1 - D_t(X_i^{s \to t}))].
\end{aligned}
\tag{1}
$$

The source-domain adversarial loss $\mathcal{L}_{gan}^s$ is defined similarly based on $T_s$, $D_s$ and $X_i^{s'}$, and the consistency loss is:

$$
\begin{aligned}
\mathcal{L}_{cyc}(T_s, T_t) =& \mathbb{E}_{X_i^s \sim \mathcal{D}_s}[\|T_s(T_t(X_i^s)) - X_i^s\|_1] \\
&+ \mathbb{E}_{X_j^t \sim \mathcal{D}_t}[\|T_t(T_s(X_j^t)) - X_j^t\|_1].
\end{aligned}
\tag{2}
$$

Following in the footsteps of CycleGAN, we utilize the identity mapping loss, which is:

$$
\mathcal{L}_{\text{identity}}(T_s, T_t) = \mathbb{E}_{X_i^s \sim \mathcal{D}_s}[\|T_t(X_i^s) - X_i^s\|_1] + \mathbb{E}_{X_j^t \sim \mathcal{D}_t}[\|T_s(X_j^t) - X_j^t\|_1].
\tag{3}
$$

Our full objective function is:

$$
\begin{aligned}
\mathcal{L}_{full}(T_s, T_t, D_s, D_t) =& \mathcal{L}_{gan}^s(T_s, D_s) + \mathcal{L}_{gan}^t(T_t, D_t) \\
&+ \lambda_1 \mathcal{L}_{cyc}(T_s, T_t) + \lambda_2 \mathcal{L}_{\text{identity}}(T_s, T_t),
\end{aligned}
\tag{4}
$$

where $\lambda_1$ and $\lambda_2$ are weighting parameters.

## 2.2   Pseudo label by multi-domain matching

We train a segmentation network using pseudo target-domain training set $\mathcal{D}_{st} = \{(X_i^{s \to t}, Y_i^s))\}$. Then we generate pseudo-label $\bar{Y}_i^t$ by inference on Sobel-MRI

sample $X_i^t$. Due to an observed fact, the pseudo-label quality of the MRI T1 sequence is generally better than other MRI sequences. The MRI dataset should be divided into several sub-datasets, i.e. there are multi-target domains. For simplicity, we split MRI dataset $\mathcal{D}_t$ into two sub-datasets $\mathcal{D}_{t1}, \mathcal{D}_{t2}$, where $\mathcal{D}_{t1}$ have better pseudo-label than $\mathcal{D}_{t2}$. Now, we hope to build a registration problem to adapt the label of $\mathcal{D}_{t1}$ to $\mathcal{D}_{t2}$. A simple way to match a moving image $X_j^{t1}$ and a fixed image $X_i^{t2}$ is to select the minimum value of two sample's volume differences.

$$\min_{\mathbf{u} \in \mathcal{D}_{t1}} \|vol(X_i^{t2}) - vol(u)\|_1 \tag{5}$$

A common way to model the deformable registration problem is to consider the minimization of an energy functional (6) over the set of plausible deformations $\phi$,

$$\mathcal{J}(\boldsymbol{\phi}; q_S, q_T) = \underbrace{\mathrm{D}(\boldsymbol{\phi}, q_S, q_T)}_{\text{Data term}} + \underbrace{\mathcal{R}(\boldsymbol{\phi})}_{\text{Regularizer}} \tag{6}$$

where $\mathrm{D}(.,.)$ is the data term that measures the discrepancy between the deformed shape $\phi.q_S$ and the target shape $q_T$. The term $\mathcal{R}(.)$ plays the role of a regularizer and thus controls the plausibility of the solution $\phi^*$. In this work, the deformations $\phi$ is an affine transform, $\mathrm{D}(A, B) = \|A - B\|_2$, and $\mathcal{R}(.) = 0$. The solution $\phi^*$ can transfer the pseudo-label of $q_S$ to $q_T$. The updated dataset still denotes $\mathcal{D}_t$.

### 2.3 Self-training

We use nnU-Net to train a segmentation model. As the trained nnU-Net may not perform well in all the images of $\mathcal{D}_t$ by using pseudo-labels, some unreliable pseudo labels may harm the training of small nnU-Net. We employ a simple method from [8] to filter the unreliable pseudo labels based on the stability during different training iterations. The method assumes that the generated pseudo labels should be stable during iterative training. If some pseudo labels vary greatly in different iterations, it indicates that the model is very uncertain about these pseudo labels. We should not use them for training. We calculate the uncertainty of pseudo labels using the following equation:

$$u = \frac{1}{K-1} \sum_{i=2}^{K} \frac{SUM(y_i \neq y_{i-1})}{SUM(y_i > 0)} \tag{7}$$

where $u$ is the uncertainty and $K$ is the total number of iterations, $y_i$ is the pseudo label generated in iteration $i$. Further, we ensemble the pseudo-label of selected cases by doing a union operation.

### 2.4 Speeding inference

In order to improve inference speed and reduce resource consumption, we use a small-size network structure in reference [8]. And we change the default resampling function and order, which effectively speeds up the inference. The setup of

Table 1: Network architecture and inference process.

| Channels in the first stage | 16 |
| --- | --- |
| Convolution number per stage | 2 |
| Patch size | 40×224×192 |
| Downsampling times | 4 |
| inference process | (Sigmoid, Threshold, Resample) |
| Deep supervision | True |

network architecture and inference process are presented in Table 1. Comparison of different strategy settings in Table 2.

Table 2: Comparison of different strategy settings. The order of axes of input patch size and spacing is (z,y,x).

| Settings | Default | Small |
| --- | --- | --- |
| Channels in the first stage | 32 | 16 |
| Convolution number per stage | 2 | 2 |
| Patch size | 40×224×192 | 40×224×192 |
| Downsampling times | 5 | 4 |
| Input spacing | (2.5, 0.81, 0.81) | (2.5, 0.81, 0.81) |

### 2.5   Data processing

**Preprocessing** For image prepossessing, all of our settings follow the default nnU-NetV2.

- Statistical analysis is conducted on data pertaining to volume spacing and foreground intensity.
- Images are clipped at the 0.5 and 99.5 percentiles of foreground voxels.
- All images are normalized through the subtraction of the mean and division by the standard deviation.
- The volume is then resampled to a target spacing of (2.5, 0.81, 0.81).

**Post-processing** We do not perform any post-processing, such as connected component analysis or testing time augmentation, during our inference.

## 3   Experiments

### 3.1   Dataset and evaluation measures

The training dataset is curated from more than 30 medical centers under the license permission, including TCIA [2], LiTS [1], MSD [22], KiTS [6,7], autoPET [5,4], AMOS [10], LLD-MMRI [12], TotalSegmentator [24], and AbdomenCT-1K [19], and past FLARE Challenges [16,17,18]. The training set includes 2050

abdomen CT scans and over 4000 MRI scans. The validation and testing sets include 110 and 300 MRI scans, respectively, which cover various MRI sequences, such as T1, T2, DWI, and so on. The organ annotation process used ITK-SNAP [29], nnU-Net [9], MedSAM [14], and Slicer Plugins [3,15].

The evaluation metrics encompass two accuracy measures—Dice Similarity Coefficient (DSC) and Normalized Surface Dice (NSD)—alongside two efficiency measures—running time and area under the GPU memory-time curve. These metrics collectively contribute to the ranking computation. Furthermore, the running time and GPU memory consumption are considered within tolerances of 15 seconds and 4 GB, respectively.

## 3.2  Implementation details

**Environment settings** The development environments and requirements are presented in Table 3.

Table 3: Development environments and requirements.

| System | Ubuntu 20.04.5 LTS |
|---|---|
| CPU | Intel(R) Xeon(R) Gold 6354 CPU @ 3.00GHz |
| RAM | 16×4GB; 1600MT/s |
| GPU (number and type) | 1 × NVIDIA A100 40G |
| CUDA version | 11.7 |
| Programming language | Python 3.10.11 |
| Deep learning framework | Pytorch 2.0.0, torchvision 0.2.2 |
| Specific dependencies | nnU-Net 2.0 |

Table 4: Training protocols.

| Network initialization | "He" normal initialization |
|---|---|
| Batch size | 2 |
| Patch size | 40×224×192 |
| Total epochs | 1000 |
| Optimizer | SGD with nesterov momentum ($\mu$ =0.99) |
| Initial learning rate (lr) | 0.01 |
| Lr decay schedule | Poly learning rate policy: $(1 - epoch/1000)^{0.9}$ |
| Training time | 10 hours |
| Loss function | Cross entropy loss and dice loss |

**Training protocols** The training protocols of the small nnU-Net are listed in Table 4. For the training set, We select 300 cases from the unlabeled MRI images

and 100 cases from labeled CT images. We use the summation between Dice loss and cross-entropy loss because compound loss functions have been proven to be robust in various medical image segmentation tasks [13]. We used the pseudo labels generated by the FLARE22 winning algorithm [8] and the best-accuracy-algorithm [23] as labels of CT. We employ the same data augmentation as the default setting of nnU-Net, which includes additive brightness, gamma, rotation, scaling, and elastic deformation on the fly during training. During inference, the model does not perform test time augmentation (TTA) of flipping. The patch sampling strategy is foreground over-sampling. Finally, we choose the model that obtains the fast and best accuracy on the validation set.

## 4    Results and discussion

Table 5: Quantitative evaluation results in terms of DSC(%) and NSD(%).

| Target | Validation | |
|---|---|---|
| | DSC(%) | NSD(%) |
| Liver | 86.97 ± 6.480 | 78.74 ± 10.44 |
| Right kidney | 84.79 ± 11.42 | 83.32 ± 13.45 |
| Spleen | 82.62 ± 18.05 | 79.54 ± 20.46 |
| Pancreas | 44.28 ± 20.63 | 56.44 ± 24.83 |
| Aorta | 79.77 ± 12.10 | 83.17 ± 14.91 |
| Inferior vena cava | 57.49 ± 15.87 | 56.16 ± 15.19 |
| Right adrenal gland | 38.16 ± 17.54 | 53.63 ± 23.91 |
| Left adrenal gland | 47.13 ± 18.12 | 61.88 ± 21.67 |
| Gallbladder | 46.71 ± 29.53 | 35.55 ± 28.70 |
| Esophagus | 35.33 ± 18.51 | 46.90 ± 23.22 |
| Stomach | 60.08 ± 17.51 | 58.93 ± 18.67 |
| Duodenum | 38.68 ± 17.29 | 61.24 ± 21.90 |
| Left kidney | 84.89 ± 16.62 | 84.55 ± 17.00 |
| Average | 60.53 ± 16.90 | 64.62 ± 19.57 |

### 4.1    Quantitative results on validation set

In Table 5, we report the DSC and NSD of the final docker commit results. The average of the 110 public validation achieve a DSC of about 0.6053 and an NSD of 0.6462. In general, large organs such as the liver, spleen, kidney and stomach have high accuracy. However, the accurate identification of small and complex objects, such as the pancreas, the adrenal glands, and the duodenum, poses a huge challenge. It requires more attention, especially when dealing with extremely small and blurred boundaries.

In Table 7, we have done a careful ablation analysis of each module and can see that training the model using only CT data is the least effective. After using

Table 6: Ablation studies of online validation quantitative evaluation results in terms of DSC(%) and NSD(%). DA is the results of using fake MRI generated by ASIT, and MDM represent to refine pseudo-labels by multi-domain matching.

| Target | w/o DA | | w/ DA | | w/ DA + MDM | |
|---|---|---|---|---|---|---|
| | DSC(%) | NSD(%) | DSC(%) | NSD (%) | DSC(%) | NSD (%) |
| Liver | 17.56 | 13.79 | 53.68 | 42.37 | 86.63 | 78.08 |
| Right kidney | 13.44 | 12.12 | 29.96 | 29.96 | 66.05 | 62.38 |
| Spleen | 5.990 | 4.290 | 46.40 | 39.70 | 79.41 | 73.01 |
| Pancreas | 11.13 | 11.39 | 12.13 | 17.02 | 38.64 | 48.73 |
| Aorta | 37.52 | 37.27 | 58.94 | 61.97 | 80.91 | 84.84 |
| Inferior vena cava | 16.05 | 15.43 | 28.28 | 26.99 | 54.29 | 54.00 |
| Right adrenal gland | 4.590 | 5.590 | 11.75 | 17.94 | 24.55 | 37.36 |
| Left adrenal gland | 10.26 | 11.39 | 10.91 | 16.56 | 24.45 | 33.98 |
| Gallbladder | 16.52 | 11.66 | 30.38 | 25.55 | 34.79 | 30.91 |
| Esophagus | 17.22 | 21.53 | 27.92 | 44.14 | 49.61 | 71.56 |
| Stomach | 13.21 | 8.810 | 34.61 | 35.92 | 59.55 | 60.62 |
| Duodenum | 15.83 | 20.06 | 10.45 | 18.34 | 23.78 | 40.18 |
| Left kidney | 19.85 | 17.66 | 28.46 | 28.65 | 65.52 | 61.75 |
| Average | 15.32 | 14.69 | 29.53 | 31.16 | 52.93 | 56.72 |

Table 7: Each module's quantitative evaluation results in terms of DSC(%) and NSD(%). ASIT is the result of using fake MRI for training on Sobel filter images, and MDM represent to refine pseudo-labels by multi-domain matching. PLF indicates that pseudo-label filtering was used.

| Module | Validation | |
|---|---|---|
| | DSC(%) | NSD(%) |
| Only CT | 15.32 | 14.69 |
| CycleGAN | 20.01 | 16.40 |
| ASIT | 29.53 | 31.16 |
| ASIT + MDM | 52.93 | 56.72 |
| ASIT + MDM + PLF | 60.53 | 64.62 |

the fake MR data generated by CycleGAN, the effect is improved. However, the generation in the anatomical space after the application of Sobel filtering gives better results. We report the validation results of the model without DA, with ASIT, and with ASIT + MDM in Table 6. The model using ASIT resulted in an increase of the DSC from 0.1532 to 0.2953. Additionally, MDM alone increased the DSC by approximately 0.23. Finally, pseudo-label filtering (PLF) further increased the DSC to 60.53. These results illustrate that it is difficult to achieve high performance for multiple target domains using only image translation.

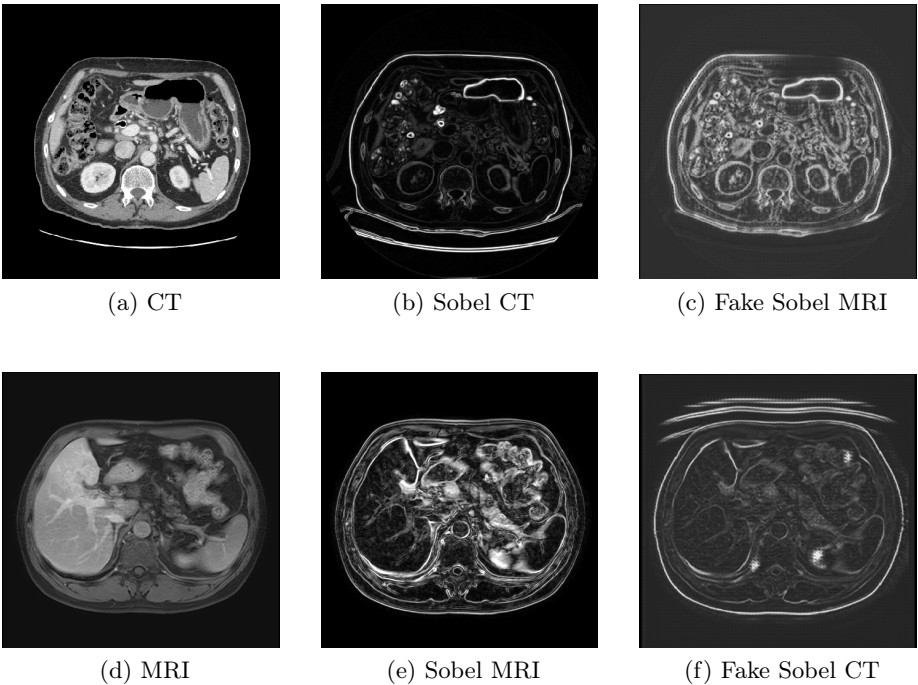

|         |           |               |
|:-------:|:---------:|:-------------:|
| (a) CT  | (b) Sobel CT | (c) Fake Sobel MRI |
| (d) MRI | (e) Sobel MRI | (f) Fake Sobel CT |

Fig. 2: 3D visualization of ASIT on CT and MRI scans.

## 4.2   Qualitative results on validation set

Figure 3 presents easy and difficult validation set examples for segmentation. Promising results were observed for Case #amos_8179 and Case #amos_0546, but the segmentation of Case #amos_7324 and Case#amos_0530 was poor. Some lesions, such as tumors, cause network segmentation errors. More importantly, the pseudo-labels generated by image translation are of poorer quality for small organs, which results in a much lower average segmentation performance.

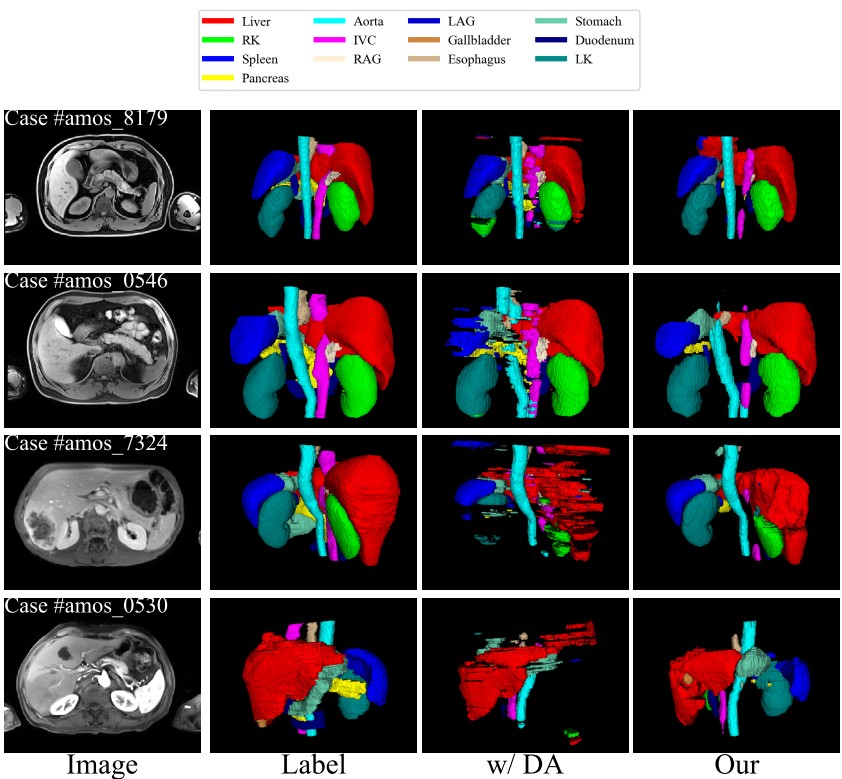

Fig. 3: 3D visualization on two easy cases (Case #amos_8179 with DSC of 0.64 and Case #amos_0546 with DSC of 0.60) and two hard cases (Case #amos_7324 with DSC of 0.45 and Case #amos_0530 with DSC of 0.48).

Table 8: Quantitative evaluation of segmentation efficiency in terms of the running them and GPU memory consumption on final testing set. Total GPU denotes the area under GPU Memory-Time curve. Evaluation GPU platform: NVIDIA QUADRO RTX5000 (16G).

| Type of statistics | Running Time (s) | Total GPU (MB) |
|---|---|---|
| 25th percentile value | 18.3 | 967283.7 |
| 50th percentile value | 19.5 | 1035840.0 |
| 75th percentile value | 21.8 | 967283.7 |
| Average value | $20.7 \pm 5.4$ | $1107332.6 \pm 316456.3$ |

Figure 2 shows the picture generated using ASIT, which is effective in shrinking the inter-domain gap.

### 4.3   Segmentation efficiency results on final testing set

In Table 8, we report the results of the quantitative evaluation of the running efficiency on the final testing set. The average inference time and GPU occupied memory are 20.7 seconds and 1107332.6 MB.

### 4.4   Results on final testing set

Table 9: Quantitative evaluation results in terms of DSC(%) and NSD(%).

| Target | Testing | |
|---|---|---|
| | DSC(%) | NSD(%) |
| 25th percentile value | 19.8 | 17.0 |
| 50th percentile value | 40.0 | 36.0 |
| 75th percentile value | 53.6 | 53.8 |
| Average value | $37.4 \pm 19.5$ | $36.3 \pm 21.3$ |

In Table 9, we report the DSC and NSD of the final docker commit results on testing set. The average of the final testing set achieve a DSC of about 37.4% and an NSD of 36.3%. The metric values on the testing set have a large decrease than on the validation set, which indicates that there is a gap between the data distribution of the testing and validation sets, and it is a challenging task to build a model with excellent generalizability.

### 4.5   Limitation and future work

This work have several limitations as follows. First, there are many ways to further improve the network inference process, such as a more efficient sliding window. Second, the challenge provided more than 3000 MRI cases, but we only utilized 300 cases and did not adequately utilize the data. Third, we perform image style translation to obtain pseudo-labeling in the decomposed space, but the quality is poor for organs with complex shapes and smaller sizes. How to effectively maintain the morphology of these organs deserves further exploration.

## 5   Conclusion

In this paper, we present a framework that combines unsupervised domain adaptation, registration and pseudo-labeling learning, which is effective and flexible for a variety of situations. In addition, we use a small nnU-Net and improve the inference process, effectively reducing its required computational resources

and inference time. Because the amount of data used in training is small, performance on the full data will be explored in the future. The approach in this paper will be a good baseline result for exploring unsupervised domain adaptation.

**Acknowledgements** The authors of this paper declare that the segmentation method they implemented for participation in the FLARE 2024 challenge has not used any pre-trained models nor additional datasets other than those provided by the organizers. The proposed solution is fully automatic without any manual intervention. We thank all data owners for making the CT scans publicly available and CodaLab [27] for hosting the challenge platform.

## Disclosure of Interests

The authors declare no competing interests.

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
