# OpenReview forum: "A Novel Pseudo Label-based Unsupervised Multiple Target Domain Adaptation Framework for Abdominal Organ Segmentation"
_MICCAI.org/2024/Challenge/FLARE — Submitted to FLARE 2024_

### Official Review · Reviewer_PJsx · 2025-01-23

**Rating:** 8
**Confidence:** 5

**Review:**

This paper presents a framework that combines unsupervised domain adaptation, registration, and pseudo-label learning to effectively and flexibly adapt to multiple target domains, resulting in a DSC of 0.6053 and a NSD of 0.6462 on 13 abdominal organs.

1. Fig. 1 is too brief. Providing additional details would enhance clarity and understanding.

2. It would be beneficial to include some visualization results of image translation for qualitative analysis.

Despite these minor suggestions for improvement, the paper is well-writen and valuable.

---

> ### Author Response · Authors · 2025-03-30
>
> Thank you for your suggestions, we have made the following changes to the paper based on the comments:
> 1. We redrew the method flowchart fig.1 to make it clearer and easier to understand.
> 2. We have added the fig.2 to show the results of image translation.

---

### Official Review · Reviewer_5GmQ · 2025-01-27
**This work proposed a UDA framework for multi-organ segmentation based on cycleGAN, multi-domain matching and self-training.**

**Rating:** 6
**Confidence:** 5

**Review:**

1. Fig 1 is unclear and needs modification and more clarification
2. Not sure of the effect of self-training, the effect of unreliable pseudo-label filtering, or the effect of the Sobel filter.
3. Lack of comparison with state-of-art methods. E.g., other UDA methods
4. Lack of visualization of image transfer, pseudo label matching
5. The visualization of segmentation results does not adhere to the official requirements. It is unclear which color corresponds to which organ.
6. Lack of the total loss function for module ASIT

---

> ### Author Response · Authors · 2025-03-30
>
> Thank you for your suggestions, we have made the following changes to the paper based on the comments:
> 1. We redrew the method flowchart fig.1 to make it clearer and easier to understand.
> 2. In Table 7, we have described each module effect, showing that each has a positive influence.
> 3. We added CycleGAN’s results to show our method’s advantage in Table 7.
> 4. We have added the fig.2 to show the results of image translation.
> 5. We revised the fig. 3 to add a legend.
> 6. We have added the total loss function eq.4.

---

### Official Review · Reviewer_UioN · 2025-03-02
**Improve the figures and tables**

**Rating:** 8
**Confidence:** 5

**Review:**

The paper is well-written overall, and the proposed method aligns with the competition goal. Please address the following minors:
Fig. 1. Add more explanations to the network

Table 5. Make the numbers aligned across different rows

Fill out the `Disclosure of Interests` section

---

> ### Author Response · Authors · 2025-03-30
>
> Thank you for your comments, we have made the following changes to the paper based on the comments:
> 1. We redrew the method flowchart to make it clearer and easier to understand.
> 2. We revise all Table to align the numbers.
> 3. We have completed section “Disclosure of Interests” .

---

### Decision · Program_Chairs · 2025-03-20

**Decision:**

Accept

**Comment:**

Please carefully address the reviewers' comments in the revision.